# GLIMO: Grounding Large Language Models With Imperfect World Models

## Abstract

Despite a widespread success in various applications, large language models (LLMs) often stumble when tackling basic physical reasoning or executing robotics tasks, due to a lack of direct experience with the physical nuances of the real world. To address these issues, we propose a Grounding Large language model with Imperfect world MOdel (GLIMO), which utilizes proxy world models such as simulators to collect and synthesize trining data. GLIMO incorporates an LLM agent-based data generator to automatically create high-quality and diverse instruction datasets. The generator includes an iterative self-refining module for temporally consistent experience sampling, a diverse set of question-answering instruction seeds, and a retrieval-augmented generation module for reflecting on prior experiences. Comprehensive experiments show that our approach improve the performance of strong open-source LLMs like LLaMA-3 with a performance boost of 2.04 $\times$, 1.54 $\times$, and 1.82 $\times$ across three different benchmarks, respectively. The performance is able to compete with or surpass their larger counterparts such as GPT-4.

## 1 Introduction

Recent advances in Large Language Models (LLMs) is bringing great transformation in various robotics applications, such as self-driving cars (Mao et al. (2023)), autonomous drones (Vemprala et al. (2023)), robotics manipulation (Liang et al. (2022)). LLMs are able to enhance robots with rich common-sense knowledge and complex planning capabilities. However, LLM needs to be physically grounded in reality, which includes understanding of the environment dynamics, the task-related constraints, and the consequences of its actions (Gao et al. (2023); Rana et al. (2023)).

Many previous works in robot learning heavily rely on prompting, such as (1) decomposing problem structures using human priors (Rana et al. (2023); Liang et al. (2022)), (2) self-refinement (Zhang et al. (2023); Wang et al. (2023a)) and (3) external tools (Mao et al. (2023)). This approach does not alter the weights of the model, instead relying on the pretrained knowledge of the LLMs. However, LLMs are trained with text corpus, lacking the understanding of the fine-grained semantics in physical environments. It also suffers from hallucination problems (Rawte et al. (2023)), and difficulties with understanding time-aware actions (Dhingra et al. (2021)). Moreover, the "heavy prompting" approach often proves effective only in small-scale environments, like a predefined room with fixed sets of objects and available actions (Rana et al. (2023)). The prompts we use may easily overfit to these small-scale environments, making the evaluation results less applicable to open-ended real-world robotic tasks.

To overcome those limitations, instruction tuning (Wei et al. (2021)) was proposed as a powerful solution by leveraging a question-answering data to finetune the LLMs and learn new capabilities (Zhou et al. (2023)). Studies also show that instruction tuning can mitigate hallucination (Ouyang et al. (2022)), enhance the agent capabilities in using tools (Schick et al. (2023)), and improve understanding of the physical world (Mao et al. (2023)).

Recent work (Xiang et al. (2023)) applied instruction tuning to enhance LLM's capability in embodied tasks with a new learning paradigm, Embodied Experiences from World Model (E2WM), which samples embodied experiences from a world model and fine-tunes LLMs with instruction data derived from those experiences. However, such an approach has several limitations. First, directly sampling experiences from a perfect world model is costly and often unsafe in real-world robotics applications, such as autonomous driving. Second, such approaches are less generalizable. E2WM assumes that

Figure 1: Overview of GLIMO (Grounding Large language model with Imperfect world MOdel). Compared with previous approaches (Xiang et al. (2023); Zeng et al. (2023)) that rely on human annotation or downstream task design, GLIMO introduces an LLM agent-based framework that autonmously navigate in the imperfect world model and annotate the embodied experiences. It improves the data quality with a self refining prompting mechanism and retrival augmented generation and enhance policy exploration in favor of data diversity.

a grounded simulator is capable of easily generating instruction datasets, whereas this is difficulty with general robotics simulators that lack text descriptions. Finally, instruction tuning requires high data diversity and quality (Zhou et al. (2023)). Yet previous approaches require human design for each specific tasks. For example, E2WM requires manually designing the downstream tasks, such as counting and object path tracking, to inject task-related knowledge into LLMs. Such approaches are not automated or scalable, further limiting its applicability to general robotics tasks.

To address these issues, we propose a Grounding Large language model with Imperfect world MOdel (GLIMO), which relaxes the assumption of a perfect world model $\tau$ by leveraging proxy world models $\hat{\tau}$, such as simulators. Our approach effectively improves the performance of LLMs on $\tau$ by fine-tuning them with embodied experiences acquired from $\hat{\tau}$. One primary challenge in our approach is the distribution shift between $\hat{\tau}$ and $\tau$. To tackle this challenge, we propose to learn domain-invariant knowledge and transfer it to the target environment. To automatically generate high-quality and diverse instruction data, we further design an LLM agent-based data generator, which samples experiences from the imperfect world model and generates instruction datasets. As depicted in Figure 1, our LLM agent framework comprises three key components: (1) **An iterative self-refining module**, which interactively samples embodied experiences in a temporally consistent manner. The LLM-in-the-loop guided sampling improves the LLM's performance on frequently accessed states during inference. (2) **A rich set of question-answering instruction seed datasets** queried across different temporal contexts to enhance data diversity and quality. (3) **A retrieval-augmented generation module** for the LLM to reflect on prior collected experiences and enhance the effectiveness of future exploration.

To evaluate the efficacy of GLIMO, we fine-tune multiple open-source LLMs, including LLaMA-3 (Touvron et al. (2023)) 8B and 70B models, LLaMA-2-13B model and OPT-13B (Zhang et al. (2022)) model. As shown as in Figure 2, we assess the performance of the fine-tuned LLMs in two environments: a 2D strategic puzzle game **Agent World** similar to Tower Of Sorcerer and a **Urban Driving** environment. Our experiments demonstrate that GLIMO enables LLMs to learn domain-robust policies with 2.04 $\times$, 1.54 $\times$, and 1.82 $\times$ of improvements in all environments, despite a distribution shift between imperfect world model and world model. Our fine-tuned model beats the SoTA close source model GPT-4 by a large margin of 51.7% and 84.5% in Agent World and Urban Driving environments, respectively.

## 2 BACKGROUND AND RELATED WORK

**Robot Learning With LLMs.** LLMs are already shown to be effective across a diverse range of robotics tasks (Mao et al. (2023); Liang et al. (2022); Vemprala et al. (2023)). Many studies are leveraging large language models to do the high-level planning for the robotics tasks (Rana et al. (2023)). Other studies also show that LLM has the potential to solve low-level control problems (Mirchandani et al. (2023)), given its strong capability in modeling the sequence. Voyager (Wang et al. (2023a)) proposes an automatic curriculum that maximizes exploration and includes an iterative prompting mechanism to absorb environment feedback and mitigate execution errors. Augmenting LLMs with external tools (e.g., web search, calculators, perception modules) is another promising approach for

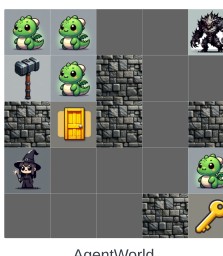 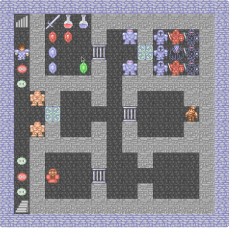 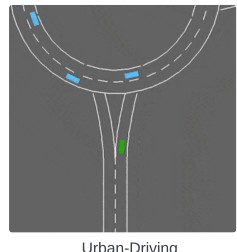 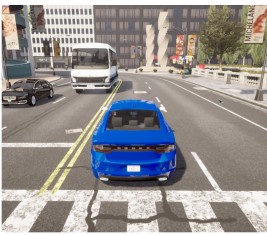

AgentWorld
(imperfect)

AgentWorld

Urban-Driving
(imperfect)

Urban-Driving

Figure 2: We collect imperfect experiences from the left imperfect world model and test it in the environment on the right. (1) The **AgentWorld** is a strategic 2D decision games, where players need to learn to improve themselves. The left figure (imperfect) is a simplified and steerable version with a smaller size and different elements and tools. (2) The **Urban-Driving** environment is based on CARLA simulator (Dosovitskiy et al. (2017)). We also build a simplified (imperfect) autonomous driving environment on the left.

enabling LLMs to interact with the real world (Mao et al. (2023)). Toolformer teaches LLM agents about tool usage through fine-tuning with demonstrations (Schick et al. (2023)).

**World Model.**  "World model" refers to a digital representation of the physical world, which can simulate real-world changes in response to actions (Ha & Schmidhuber (2018)). A large body of previous studies on machine learning and cognitive science leveraged world model to enable data-efficient learning and strong generalization in unseen scenarios (Allen et al. (2020); Battaglia et al. (2013)). Humans have an innate world model for predicting outcomes of actions. Recent studies develop these models in LLMs for enhanced reasoning. Xiang et al. (2023) fine-tuned LLMs with collected experiences in a grounded physics engine, VirtualHome (Puig et al. (2018)), to mirror real conditions. However, those approach is infeasible in real-world robotics tasks due to its high-cost and safety concerns in directly collecting embodied experiences from the real world. Instead, our approach mitigates these issues by enabling learning from imperfect world model, such as simulators.

**Language Model Grounding.**  Recent research has focused on integrating language models (LMs) with world models to improve task performance. Techniques vary from prompting (Wang et al. (2023a)) and fine-tuning through supervised learning (Wei et al. (2021)) to reinforcement learning (Ouyang et al. (2022)). Previous works (Xiang et al. (2023)) show that LLMs can improve their capabilities by fine-tuning on embodied experiences collected from world models. This process involves adjusting the model's parameters to better reflect the complexities and nuances of real-world interactions. By simulating scenarios within world models, LLMs can learn to predict outcomes more accurately and adapt to dynamic environments. This method has proven especially beneficial in scenarios where LLMs must interact with physical or simulated environments, enhancing their decision-making and problem-solving abilities.

## 3 METHODOLOGY

### 3.1 PROBLEM FORMULATION

LLMs, parameterized by $\theta$, learns to predict the distribution of next word $x_t$, given the preceding text sequence $x_{1:t-1}$:

$$P_\theta(x_t|x_{1:t-1}) \tag{1}$$

Large language models (LLMs) are often viewed as world models in textual space, containing extensive knowledge that can assist in decision-making within physical environments. Previous research shows that sampling embodied experiences from physical environments can help LLMs better understand those environments (Xiang et al. (2023)), which is often infeasible due to costs and safety concerns in real-world robotics tasks.

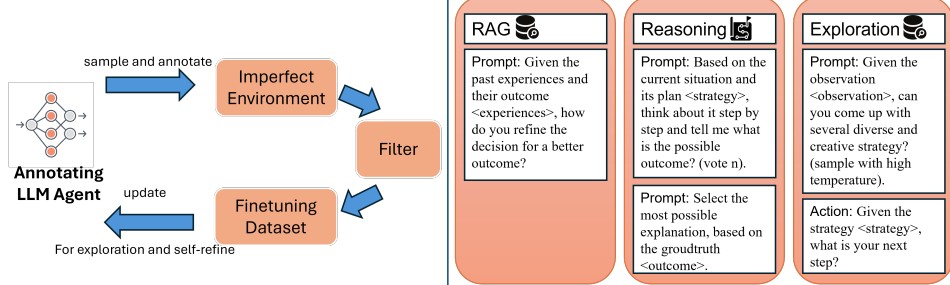

Figure 3: (1) On the left shows the LLM agent-based pipeline guiding the sampling of embodied experiences and transforms these experiences into formats suitable for instruction tuning. In each iteration, the LLM agent executes one plan evaluates whether the goal is achieved. The LLM also compares the current plan with history experiences and annotates them on different length of steps. (2) On the right shows our prompt templates to (i) self-refine with past experiences (ii) generating rationale (or explanation) for decisions, to enable chain-of-thoughts finetuning Lightman et al. (2023). (iii) to favor exploration and also enhance the diversity of our data. These templates guarantee our agents be able to produce high-quality and diverse embodied experiences.

In this paper, we propose a novel learning paradigm of grounding large language models, GLIMO, that collects embodied experiences from imperfect world models. GLIMO enables LLM to learn from sampled embodied experiences from an imperfect world model $\hat{\tau}(s_t|s_{t-1}, a)$, which is often accessible, low-costs, and interactive, in the hope of optimizing its performance in real-world (perfect) environment $\tau(s_t|s_{t-1}, a)$. Formally:

$$\min_\theta \mathbb{E}_{x \sim \hat{D}, \hat{D} \sim \hat{\tau}(s_t|s_{t-1},a)} L(y, f(x; \theta)) \tag{2}$$

$f_\theta(x)$ is the autoregressive neural language model, $x$ is the input prompts and $y$ is the corresponding label. $L$ is a cross-entropy loss that is used to train the language model $f_\theta$.

By relaxing the assumption of perfect world model, this framework facilitates a more cost-efficient and generalized learning paradigm . A typical example is the simulator, it can simulate imperfect experiences $\hat{D}$ in the virtual world efficiently, while the problems such as sim2real gaps render the synthesized data less useful, due to the simulator cannot perfectly model the real-world distribution. We conjecture and prove that LLMs are able to learn useful information from imperfect world models such as simulators.

**Perfect & Imperfect Environments.** We deploy LLM-based agents in an environment with a state distribution $s_{test}$, while the LLM is fine-tuned on a training dataset $D$, sampled from a state distribution $s_{train}$. Inevitably, the embodied experience $D_{test}$ differs from $D_{train}$ due to *distribution shifts*, which may cause LLMs to learn spurious correlations rather than the true causal structure of the environment.

To assess whether LLMs can learn policies that generalize across different distributions, we conduct a case study using two environments to validate our methods. Figure 2 shows the visualization of: (1) **Agent World**, a 2D puzzle game adapted from "The Tower of the Sorcerer," requiring strategic planning across 42 tasks; and (2) **Urban Driving**, a self-driving simulator based on CARLA (Dosovitskiy et al. (2017)), which generates varied traffic scenarios to test the LLM's ability to handle safety-critical events. The detailed description is in Appendix A.3. Each environment has an imperfect and perfect version. Appendix A.4 shows a detailed comparison of the distribution shift.

### 3.2 GENERATING GROUNDED EMBODIED EXPERIENCES WITH LLM AGENTS

Instruction tuning (Wei et al. (2021)) is an effective approach to enhance LLMs' capabilities in domain-specific problems. However, instruction tuning requires a textual question-answering format. To fit this paradigm, previous works manually design formats to extract knowledge from collected experiences, such as plan generation and object path tracking (Xiang et al. (2023)). These approaches,

which integrate human prior knowledge about what is useful in specific tasks, are still unwieldy and difficult to scale in real-world robotics tasks.

To address this, we propose to use an LLM agent-based framework to interpret embodied experiences and automatically transform it into instruction tuning format, in a scalable manner. The LLM-based agent framework keeps track of the embodied experiences in the target environment $\hat{D}$. The LLM interacts with the embodied experiences as a multi-turn dialogue, and generates grounded instruction data. Our framework let the LLM to explore the environment via a self-refining iterative prompting mechanis (Shinn et al. (2023); Wang et al. (2023a); Yao et al. (2023)).

Figure 3 illustrates the pipeline for our LLM agent-based system, designed to guide the sampling of embodied experiences and optimize them for instruction tuning. On the left side of the figure, the LLM agent iteratively executes a plan and evaluates whether the defined goal has been achieved. During each iteration, the agent not only assesses the success of the current plan but also compares it against prior experiences. This iterative approach ensures that the agent continuously learns and adapts from historical data.

We present the prompt templates that are integral to this process. These templates serve three key functions: (i) enabling self-refinement based on past experiences, (ii) generating rationales or explanations for the agent's decisions to support chain-of-thought fine-tuning Lightman et al. (2023), and (iii) encouraging exploration to enhance the diversity of the generated data. These prompt templates ensure that the LLM agents produce high-quality, diverse embodied experiences, which are essential for refining the agent's decision-making capabilities in dynamic environments. We will introduce those with more details in the remained section.

**Agent Integration.** We use text-only LLMs in this paper, as we focus on whether the textual embedding space learned by LLMs is able to transfer the dynamics from imperfect world model to perfect world model, which also helps our future study in multimodal LLMs. To generate text inputs and enable actions for LLMs, we introduce: (1) a tool module, that interacts with the environment to provide necessary information. For example, a neural object detection module, or a collision detection module. LLM agents will call the tool to acquire necessary information for the decision-making. (2) A customizable memory module that explicitly retains previous actions and observations, integrating human prior knowledge about what is necessary past information for future tasks. (3) A skill library that encodes necessary skills for LLMs to act in the physical world. The skills works as a function calling with certain format: e.g., a `lane_change(lane_id:int)` skill can be called, and we have a low-level implementation to actuate this skill. The skill library is needed to cope with continuous control task.

**Reasoning.** Reasoning is important in sampling experiences, reject trivial experiences and explore valuable directions. We use prompt engineering techniques such as Chain-of-thoughts (Wei et al. (2022)), ReAct (Yao et al. (2023)) and Reflexion (Shinn et al. (2023)) to faciliate reasoning and refine the LLM's decisions, and mitigate error accumulation. To enhance the LLMs' ability in reasoning, we use an iterative prompting method as in (Wang et al. (2023a)) to self-refine and verify proposed action sequences. By performing exploration actions iteratively, the LLM learns to self-reflect and refine its actions. Figure 3 shows a self-consistency example to generate n reasoning data and using votes to select the groundtruth Wang et al. (2023b).

### 3.3 RETRIEVAL-AUGMENTED EXPERIENCES

Imagine a game in which the player makes a single decision, choosing between option A and option B. The success rate of option A is 100%, while the success rate of option B is 80%. Without any prior knowledge, the LLMs should treat both options as equal. Prior works, such as supervised finetuning, adjust the LLMs' weights based on maximum likelihood principles. As a result, the LLM might slightly favor option A. However, in real-world decision-making, we desire a deterministic policy that always chooses option A. This example highlights the importance of reflecting on embodied experiences along with past experiences.

We introduce a RAG (Retrieval Augmented Generation) pipeline to self-reflect and self-improve prior experiences while promote exploration. As shown in Figure 3.3, our framework retrieves previous related embodied experiences, and generate instruction data based on this contexts (Lewis

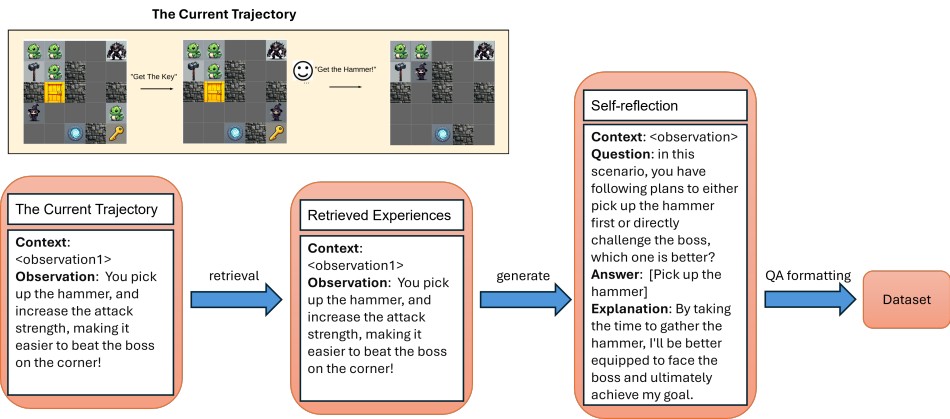

Figure 4: This figure illustrates retrieval-augmented generation involving reasoning about the feasibility of different high-level plans, such as: "Which plan is better, challenge the boss directly or get the hammer first?" This approach allows us to reflect on prior experiences, determine which is better and generate reasoning data, which can be further transformed into QA datasets with varying format, as introduced in 3.4.

et al. (2020)). We use LLM to tag the action sequence of each experience, mainly describing the strategy and outcome, for example: "firstly get the yellow keys, open the door with the key, grab the hammers". We embed the tag with SFR embedding by Salesforce( Rui Meng* (2024)) and retrieve relevant experiences. Such retrival enhance the reasoning of LLMs by providing more diverse and high-quality few-shot examples, which is particularly useful in domains where expert demonstrations are abundant. We also ask LLM to label the surprising score of the instruction data (on a scale of 1-10), which can be used as weighted sample to upweight the difficult samples, as in Xiang et al. (2023).

We also leverage domain specific models to do the retrieval. For the urban driving scenario, we used a scenario encoding model, as used in MTR [2]. The MTR transformer first transform the driving scenario into a vectorized representation, and use a transformer-based encoder to encode it into an embedding. The original MTR paper can decode the embedding to predict the future trajectory, in our work, we use it to retrieve similar scenarios. One future exploration to leverage multimodal embeddings to retrieve experiences with vision modalities [3].

## 3.4 ANNOTATION GENERATION

Our pipeline automatically transform embodied experiences to grounded instruction data formats. Motivated by self-instruct (Wang et al. (2022)), we begins with a small set of task templates. Random tasks are sampled from this pool and used to prompt the LLM to generate both new instructions and corresponding instances. Low-quality or similar generations are then filtered out, and the remaining high-quality instances are added back to the initial repository of tasks. Such self-refining and bootstrapping process improve the diversity and quality of the generated data. They also constantly receiving the feedback from the simulators to correct possible errors.

Notable, our task pool includes the following tasks which significantly improve the finetuned model: (1) **Rationale Enhancement.** Incorporating explanations or rationales has been shown to improve LLM performance, as indicated by previous studies (Lampinen et al. (2022)). By understanding the reasoning behind decisions, LLMs can form more robust strategies that better mimic human-like reasoning. (2) **Counterfactual Reasoning** involves training LMs to consider alternative outcomes or scenarios that didn't occur but could have, under different circumstances. By engaging in this type of hypothetical reasoning, LMs can develop a deeper understanding of causality and potential consequences in physical environments, leading to more informed and effective decision-making. (3) **Episodic Memory Question Answering** (Datta et al. (2022)) is a question-answering format that queries pre-recorded embodied experiences to assess the spatial and temporal understanding of a scenario. For example, it might ask, *"Where is the altar you encountered most recently?"*.

We also annotate the grounded instruction dataset by querying on different temporal scale. For example, Figure 3.3 shows the comparison of two **high-level plans**: getting the hammer and attack the boss. Those actions are carried out with a sequence of low-level actions of up/down/left/right movement. We also design different questions on a shorter temporal scale, e.g. "*Is it able to access the yellow key directly?*"

**Filtering.** The purpose of the filtering is to encourage diverse and high-quality data samples, which is critical for the performance of instruction tuning Ouyang et al. (2022); Wang et al. (2022). We find two approaches effective in our experiments: Ask LLM to determine if the new data is novel with chain-of-thoughts prompting, according to the tagging as mentioned in 3.3. We also leverage self-consistency to generate multiple samples to evaluate the novelty, and take the majority of the voting results. Use domain specific model or generative model to embed the observation, and filter the samples that are in distribution. For the urban-driving scenario, we use a variational autoencoder and leverage the reconstruction loss to evict easy samples, a low reconstruction loss means the data is in-distribution, and easy to predict. This is actually similar to the idea of hard negative data mining.

**Training.** We take a two-stage finetuning approach: (1) in the first stage, we improve the instruction following capability of our base model as in (Liu et al. (2024)), Chat-QA instruction dataset also specializes in understanding tabular data and text retrieval, which we deem very useful in our embodied setup. (2) in the second stage, we finetune our model with our grounded embodied experiences. To save computation resources and avoid overfitting, we use LoRA techniques (Hu et al. (2021)) to finetune LLMs such as LLaMA-3 (Touvron et al. (2023)) and OPT-13B (Zhang et al. (2022)).

## 4 EXPERIMENTS

In this section we present the training and evaluation details of our experiments.

### 4.1 TRAINING DETAILS AND BASELINES

**Experimental Details.** We test our methods mainly on the powerful LLaMA-3-8B, LLaMA-3-70B Touvron et al. (2023), and OPT-13B models Zhang et al. (2022). We use Q-LoRA finetuning techniques Dettmers et al. (2023) with 8-bit quantization, due to limited computation resources. We finetune our models on $4 \times$ NVIDIA A6000 and $8 \times$ NVIDIA H100 clusters (For 70B models mainly). For the annotating LLMs agent, we leverage the OpenAI `gpt-4-0314` APIs and Gemini Pro for embodied experiences annotation. When finetuning the LLM, the training process can be unstable, leading LLMs to lose its general language capability Luo et al. (2024a). Such phenomenon, called catastrophic forgetting, is often mitigated by a Kullback–Leibler divergence regularization term, to avoid the language model weights from diverging too much from the original model Xiang et al. (2023); Rafailov et al. (2023).

**Hyperparameters.** For LLaMA-3 models, we use a learning rate of $6 \times 10^{-5}$. For OPT models, we use a learning rate of $2.5 \times 10^{-5}$. Our approach requires about 19 and 45 hours to train LLaMA-3-8B, 70B models and 23 hours to train OPT-13B. We used a rank of 128 and a coefficient of 256 for LoRA's hyperparameters.

**Baselines.** We offer a model-agnostic LLM agent framework, built using LangChains (lan). Primarily, we utilize ReAct (Yao et al. (2023)) and provide essential tools, such as object detection for **urban driving** environments. Given the model-agnostic nature of the LLM agent, we use the base LLM as the baseline and integrate our fine-tuned models to compare performance. For each task, the LLM agent is run 10 times, and we calculate the mean performance/accuracy.

### 4.2 FINETUNED BASELINES.

We introduce a baseline that fine-tunes LLMs using data sampled from a perfect world model, as shown in Figure 5. A ratio of 0 indicates fine-tuning exclusively with data from the imperfect world model, while a ratio of 1 represents fine-tuning solely with data from the perfect world model. The overall dataset size remains constant to eliminate any influence from varying dataset sizes.

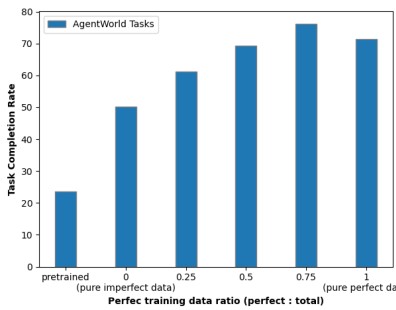 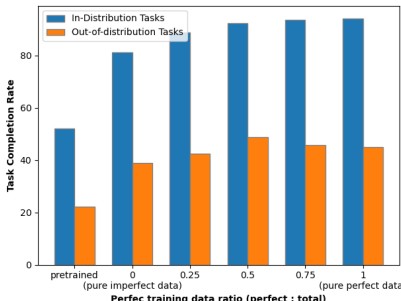

Figure 5: We compare the performance (success rate) our framework with varied ratio of imperfect training data, ranging from 0 to 1. (1) the left figure shows the performance of AgentWorld environment. (2) the right figure shows the performance of Urban-driving environment, we also evaluate its generalization capability in OOD scenarios.

Our key insights are as follows: (1) Increasing the proportion of perfect world model data enhances model performance, but the performance gains diminish rapidly beyond a ratio of 0.25. (2) Training with data from the imperfect world model actually improves the generalization capability of LLMs, particularly in out-of-distribution (OOD) scenarios. This suggests that data from the imperfect world model can serve as an effective "regularization" technique to mitigate overfitting, a common challenge in LLM supervised finetuning (Luo et al. (2024b)).

## 4.3 BASE MODEL AS BASELINE

In this section we compare our performance with base model. Our evaluation focuses on testing the task completion rate within the environment. Additionally, we use question answering to assess whether LLMs understand the fine-grained information relevant to their actions. To evaluate the model's capability to extrapolate and apply knowledge in unfamiliar scenarios, we use out-of-domain (OOD) question answering data. We also study the exploration capability with an increasing number of attempts, as shown in Figure 7.

**Task Completion.** We measure the task completion percentage in those environments. The tasks are deemed successful if it achieves certain goals such as "beat the boss monsters", "enter the next level", and "reach the destination lane safely". We report the task completion performance in Figure 6 (a), on finetuned and original LLaMA-3 8B, 70B and OPT-13B. As a comparison, we also report the performance of the GPT-4 model, as the current state-of-the-art closed source LLM. We can see that our finetuned models achieve great improvement from base models, and the finetuned LLaMA-3 models are able to beat the GPT-4 model, demonstrating the effectiveness of our approach.

**Environment Understanding.** To evaluate if the LLMs are able to understand the fine-grained dynamics of the environment (such as if lane changing to the right is safe), we also curate question answering datasets (1000 QA data for each perfect environment) from those collected embodied experiences. We list those question format in Appendix A.2. The question answering accuracy is shown in Figure 6 (b). We also show its accuracy for out-of-distribution scenarios (OOD) to showcase zero-shot generalization capability. We can see that except OPT-13B models, our approach consistently achieve considerable improvements in environment understanding. The LLaMA-8B and 70B models are able to beat GPT-4 in **Agent-QA** and **Driving-QA** datasets.

**Language Modeling.** We also evaluate the finetuned model's perplexity on the general language modeling tasks, as a metric about catastrophic forgetting. Table 1 shows that our finetuned model has a slightly increased perplexity value on PILE dataset (Gao et al. (2021)), demonstrating the perservation of generality and language modeling capabilities of LLMs.

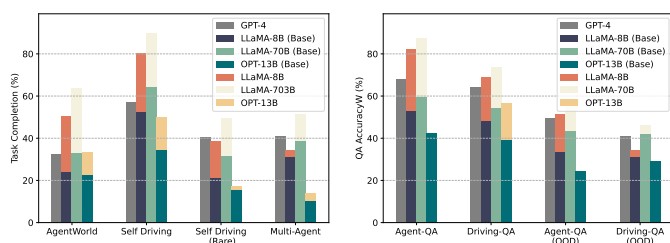

Figure 6: Performance and accuracy of our framework evaluated in two environments.

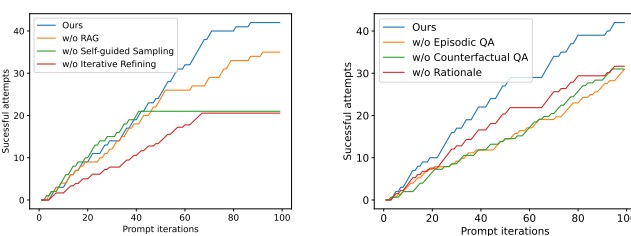

Figure 7: Ablation study on annotation LLM agent design choice and seed instruction data format.

## 4.4 ABLATION STUDY

We ablate the different design choices and different QA format in Figure 7 to highlight their individual contribution. We find that self-guided sampling and iterative refining significantly improve the quality of embodied experiences. Besides, the counterfactual QA and episodic QA improves the reasoning of the finetuned LLMs by exposing novel states to the collected experiences.

**Scaling Performance.** We report the task completion of LLaMA-2, from 7B, 13B, to 70B on the validation QA datasets in Table 1. We see that the performance consistently improve as the parameter number increases, showing the potential of our approach with scaling computation budgets.

Table 2: Results of using different LLMs as the annotating agents. We evaluate the model in two environments and report the task completion percentages.

|  | Gemini-Pro | GPT-4 | LLaMA-3 | |
|---|---|---|---|---|
|  |  |  | 8B | 70B |
| Agent World | 58.5 | **78.45** | 51.2 | 61.2 |
| Self-driving | 41.12 | **50.32** | 37.8 | 43.4 |
| Multi-agent | 39.2 | **45.4** | 38.1 | 45.6 |

## 4.5 DATA QUALITY STUDY

Because we do not train the annotating LLMs, we also study how different LLM models affect the quality of instruction datasets, as shown in Table 2. From LLaMA-3 8B to 70B, we observe that the data quality consistently improves with growing capability of the LLMs. GPT-4 still achieves an optimal data annotation quality among those LLMs.

| Model | LLaMA-2-7B | LLaMA-2-13B | LLaMA-2-70B |
|---|---|---|---|
| Task Completion(%) | 68.5% | 72.9% | **83.5%** |
| Perplexity | 4.37 | 4.24 | **3.98** |
| Original Perplexity | 4.62 | 4.55 | 4.04 |

Table 1: Results of base models with various parameters. We report task completion percentages and the perplexity of the outputs on autonomous driving scenarios and **DrivingQA.**

## 5 LIMITATIONS

Due to the limitations of conducting real-world testing in academia, including insufficient resources and costs, our approach is evaluated solely in simulated environments, such as games and self-driving simulators, which may not sufficiently generalize to complex real-world scenarios. Additionally, our LLMs currently accept only text inputs, while robotics applications require perception modalities to comprehend fine-grained environmental semantics. In future work, we aim to extend our approach to multimodal LLMs, as they can incorporate perception capabilities.

## 6 CONCLUSION

In this paper, we presented GLIMO, a novel approach to enhance the performance of large language models (LLMs) in robotics applications by leveraging imperfect world models. By using an automatic LLM annotation agent that includes an iterative self-refining module to explore the environment, a diverse set of question-answering instruction sets, and a retrieval-augmented generation module, we demonstrated that GLIMO can effectively fine-tune LLMs using embodied experiences from imperfect world models.

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

# A  APPENDIX

## A.1  ENVIRONMENT DESCRIPTION

---

**Prompt .1**     **Observation (Or the LLM input prompt), simplified for illustration**

### STATUS

The player starts at position (0, 0) on the first floor. The goal is to navigate through the tower to reach the stairs at position (4, 4) while defeating monsters (if necessary) and collecting items.
Health: 535
Attack: 12
Defense: 8
You current backpack:
YELLOW KEY: 2
BLUE KEY:1

### MAP LAYOUT (CATEGORY: DESCRIPTION ID)

Row 1: (Player: P1), (Empty: E1), (Empty: E1), (Wall: W1), (Health Potion: H1)
Row 2: (Empty: E1), (Empty: E1), (Empty: E1), (Wall: W1), (Monster: M1)
Row 3: (Empty: E1), (Empty: E1), (Monster: M2), (Empty: E1), (Empty: E1)
Row 4: (Empty: E1), (Key: K1), (Empty: E1), (Monster: M3), (Empty: E1)
Row 5: (Empty: E1), (Empty: E1), (Empty: E1), (Empty: E1), (Stairs: S1)

### DESCRIPTION IDS AND THOROUGH DESCRIPTIONS

PLAYER (P1)

- **Position:** (0, 0)
- **Description:** You are the warrior, starting from this position. Your goal is to reach the stairs at (4, 4) and progress through the tower.

---



**Observation (Or the LLM input prompt), simplified for illustration**

- **Attributes:**
  - **Health:** 100
  - **Attack:** 10
  - **Defense:** 5

EMPTY (E1)

- **Position:** Various

- **Description:** This is an empty tile with no obstacles or monsters. You can move freely through these tiles.

WALL (W1)

- **Position:** Various

- **Description:** This is a wall tile that cannot pass, you can destroy the tile with a shovel (if any).

RED DOOR (D1)

- **Position:** Various

- **Description:** This is a door tile that can be opened with a key.

HEALTH POTION (H1)

- **Position:** (0, 3)

- **Description:** A health potion that restores 20 health points when collected.

- **Effect:** +20 Health

RED KEY (K1)

- **Position:** (3, 1)

- **Description:** A red key used to unlock red doors and access other areas of the tower.

- **Effect:** Unlocks locked a red door, you lose the item once you use it.

MONSTER (M1)

- **Position:** (1, 2)

- **Description:** A low-level monster blocking your path.

- **Attributes:**
  - **Health:** 50
  -
  - **Attack:** 15
  - **Defense:** 5

MONSTER (M2)

- **Position:** (2, 2)

- **Description:** A medium-level monster that poses more danger.

- **Attributes:**
  - **Health:** 70
  -
  - **Attack:** 20
  - **Defense:** 8



---

**Observation (Or the LLM input prompt), simplified for illustration**

MONSTER (M3)

- **Position:** (3, 3)
- **Description:** A strong monster that guards valuable areas.
- **Attributes:**
  - **Health:** 100
  - 
  - **Attack:** 25
  - **Defense:** 10

STAIRS (S1)

- **Position:** (4, 4)
- **Description:** These stairs lead to the next floor of the tower. Reaching them will allow you to progress to the next level.

---

A.2    DATASET FORMAT

---

**Prompt .2**                                                                      **Agent World**

Question: Which item should you prioritize collecting to increase your defense strength?
Context: A: Blue Crystal B: Red Crystal C: Health Potion
Answer: A
Question: What should you do when you encounter a monster with higher defense strength than your attack strength?
Context: A: Engage in battle B: Search for attack crystals C: Drink a health potion
Answer: B
Question: Which action maximizes your health conservation in the early levels?
Context: A: Fighting weaker monsters first B: Donating gold to altars C: Skipping battles and collecting potions
Answer: A
Question: How do you increase your attack strength efficiently?
Context: A: Collecting attack crystals B: Drinking health potions C: Avoiding battles
Answer: A
Question: What should you do when the Orb of the Hero shows a monster will inflict high damage?
Context: A: Engage in battle B: Seek an alternate path C: Use a health potion
Answer: B
Question: Which monster should you challenge first on a new floor?
Context: A: The one with the lowest health B: The one with the highest attack strength C: The one guarding the key
Answer: A
Question: What is the best strategy for donating gold to altars?
Context: A: Donate all gold at once B: Donate small amounts regularly C: Save gold for later floors
Answer: B
Question: What is the immediate benefit of collecting a health potion?
Context: A: Increased attack strength B: Increased defense strength C: Increased health
Answer: C
Question: What should you consider before engaging a boss monster?
Context: A: Your current health and strength levels B: The number of keys you have C: The location of the nearest health potion
Answer: A
Question: How can you determine the optimal path through a floor?
Context: A: By engaging the strongest monsters first B: By avoiding all battles C: By using the Orb of the Hero to assess monster damage

---

**Agent World**

Answer: C

**Prompt .3**                                                                           **Urban-Driving**

Question: what may be the future movement of red vehicle in front?
Context: A: lane change to the left B: Going straight C: lane change to the right
Answer: A
Question: What's the last lane you are in? Answer the lane id
Context: N/A
Answer: [lane_id]
Question: Which object in the scenario is the most safety-critical to your decisions?
Context: N/A
Answer: [obj_id]
Question: Is this trajectory likely to collide with vehicle A?
Context: N/A
Answer: No
Question: What is the most probable action of the pedestrian on the right sidewalk?
Context: A: Crossing the street B: Walking along the sidewalk C: Standing still
Answer: A
Question: Which lane should you merge into to follow the planned route?
Context: A: Left lane B: Middle lane C: Right lane
Answer: B
Question: What is the likely future movement of the blue truck behind?
Context: A: Accelerating B: Decelerating C: Maintaining speed
Answer: A
Question: How should you adjust your speed given the yellow traffic light ahead?
Context: A: Speed up B: Maintain current speed C: Slow down
Answer: C
Question: What should be your immediate action when the vehicle in front suddenly brakes?
Context: A: Brake immediately B: Change lanes to the left C: Change lanes to the right
Answer: A
Question: What is the expected behavior of the cyclist on your left?
Context: A: Turning right B: Continuing straight C: Stopping
Answer: B
Question: How should you respond to the emergency vehicle approaching from behind?
Context: A: Speed up B: Maintain current speed C: Pull over to the side
Answer: C
Question: What is the likely action of the white SUV merging from the right?
Context: A: Accelerating to merge B: Decelerating to yield C: Maintaining speed
Answer: A
Question: What is the best course of action when approaching a school zone?
Context: A: Increase speed B: Maintain current speed C: Reduce speed
Answer: C
Question: What should you anticipate from the car signaling a left turn ahead?
Context: A: Immediate turn B: Decelerate and then turn C: Continue straight
Answer: B

## A.3 ENVIRONMENTS

**Agent World.** The Tower of the Sorcerer tow is a puzzle game where the player (as a warrior) takes an adventure in a 17-level tower, the player needs to explore the tower, improve itself to get ready for the final boss and save the princess. Players enhance their character's attack strength, defense, and health by collecting items, such as potions, crystals, and equipment, and by donating gold at altars to acquire experiences or strength. The game features a deterministic combat system where players and monsters take turns striking each other, with damage calculated as the difference between attack and defense strengths. This necessitates thoughtful planning and strategic battle order to efficiently progress through levels while conserving resources like health and keys. A interesting tool is *the Orb*

*of the Hero*, which provides vital information on monster capabilities and potential damage, aiding players in choosing their battles wisely.

Key to strategy in the Tower of the Sorcerer is the Orb of the Hero, which provides vital information on monster capabilities and potential damage, aiding players in choosing their battles wisely. This intricate setup makes the game an excellent platform for evaluating decision-making algorithms, as it requires advanced reasoning, meticulous planning, and ongoing strategy refinement—crucial capabilities for sophisticated AI systems.

We design a 2D puzzle game by readapting The Tower of the Sorcerer tow. The game requires the player to plan its action wisely, for example, practice on the weak monster to gain the experiences and gold, then use the gold to improve himself at the altar. LLM needs to explore the environment and come up with optimal strategies.

The input of the agent world is a 2D string array, each elements are a description of the position, including *wall*, *boss*. Since the game takes very long durations and has very diverse challenges in reasoning and planning, We redesign the original games into 42 tasks across different levels, each tasks comes with an initial states and a final goal. We also build **Agent-QA** as annotated question-answering datasets based on those tasks. We manually filter and verify the correctness of those data. Figure 1 shows an visualization of both environments.

**Urban Driving.** We build a self-driving simulator based on the *highway-env* Leurent (2018). The simulator can randomly generate different maps such as intersections, roundabouts and junctions. It simulate the behaviors using a rule-based Intelligent Driver Model (IDM) Treiber et al. (2000), with a discrete action space based on the IDM, including change lanes, accelerate and decelerate. To evaluate, we curate a list of typical unseen scenario by clustering real-world driving scenarios from Waymo dataset Ettinger et al. (2021). To test the ability to handle long-tailed events, we generate safety-critical scenarios Liu et al. (2023); Rempe et al. (2022). We curate those unseen scenarios as a **Driving-QA(OOD)** datasets. Refer to Appendix for more information.

We use CARLA simulator to render and simulate the physics and behavior of the environment. The following list shows our testing scenarios and their brief descriptions:

- **Vehicle Following**: The autonomous vehicle (AV) is following another vehicle in the same lane, trying to keep a safe distance.
- **Yielding**: Another vehicle in the neighboring lane is trying to cut in; the AV should yield to avoid crashes.
- **Cut-in**: The AV is about to cut in.
- **Intersection**: Steering at the intersection.
- **Merging Lane**: Entering the merging lane.
- **Road Curve**: The AV should avoid the road edge.
- **Occlusion**: An unexpected pedestrian or vehicle shows up from behind an occlusion.
- **Reckless Driver**: Another vehicle not following traffic rules, such as running a red light or making an illegal U-turn.

**OOD Scenario Description** Here is a list of safety-critical scenarios we used for generalization study in Figure 6 and Figure 5.

- **Erratic Merging:** A vehicle merges into your lane abruptly without signaling or at an unsafe speed.
- **Cut-In:** Another vehicle suddenly cuts into your lane, often from an adjacent lane or during heavy traffic.
- **Nudging:** A vehicle gradually moves closer to your lane without fully changing lanes, causing a potential sideswipe situation.
- **Hard Braking:** A vehicle in front of you suddenly decelerates, forcing you to react quickly to avoid a collision.

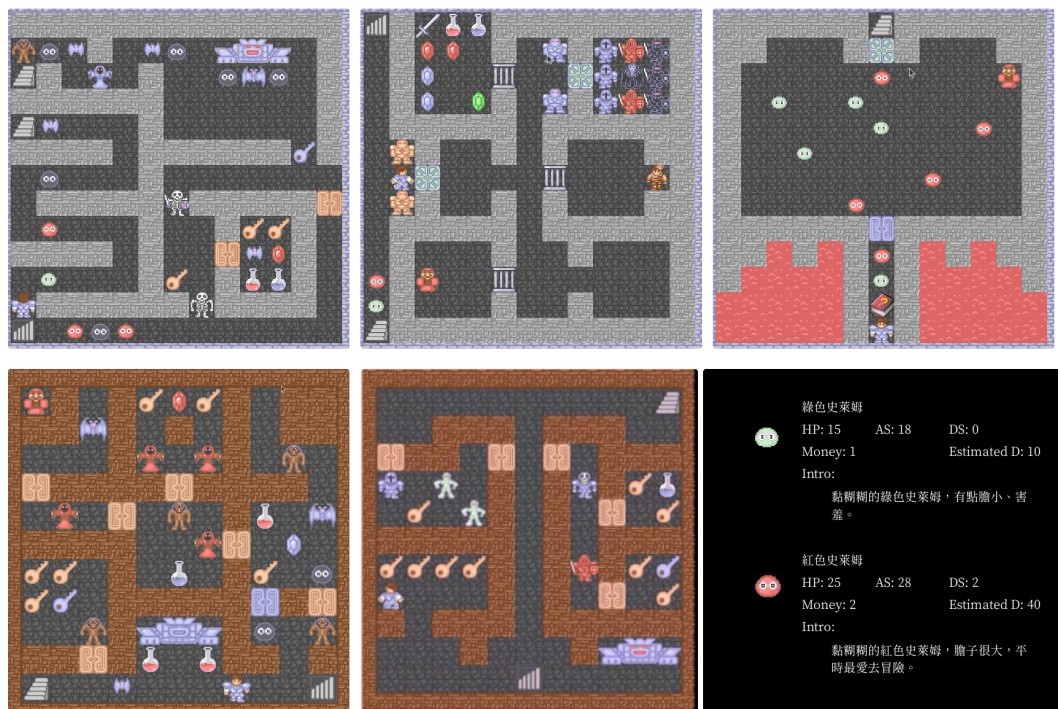

Figure 8: The AgentWorld environment includes a diverse range of tasks including challenges in exploration, tool usage, and planning. The last figure shows the *Orb of the Hero*, which shows the strength of all the monsters in current level, providing necessary informaiton for warriors to plan.

- **Lane Departure:** A vehicle unintentionally drifts out of its lane, potentially into oncoming traffic or off the road.

- **Intersection Encroachment:** A vehicle enters an intersection at the wrong time, violating right-of-way rules.

- **Pedestrian Crossing:** A pedestrian suddenly appears on the roadway, requiring immediate action to avoid an accident.

- **Unexpected Obstacle:** An object or animal suddenly appears on the road, necessitating a quick response to avoid impact.

- **Aggressive Driving:** Another driver exhibits aggressive behavior such as tailgating, speeding, or weaving through traffic.

**Multi-agent.** Previous research has already shown that LLMs often lack the Theory Of Mind Bara et al. (2021), which is the capability to explain, predict, and interpret behaviors by attributing mental states such as beliefs to oneself or the others. When applying LLMs on real-world AI tasks that involves interacting with humans, or multi-agent setting that requires cooperations, it is essential for LLMs to understand, or suitably expect others to work. We build special settings in the previous two environments that requires collaboration between agents. For example, two warriors are trapped in different rooms in **Agent World**, they need to collaborate to trigger the mechanism to open the door for each of them.

## A.4 DISTRIBUTION SHIFT BETWEEN IMPERFECT AND PERFECT ENVIRONMENT

This section describes the details of the two environment AgentWorld, in terms of their difference between imperfect world model and world model. Since we use text-only input for the LLMs, the major difference is the environment setup and dynamics.

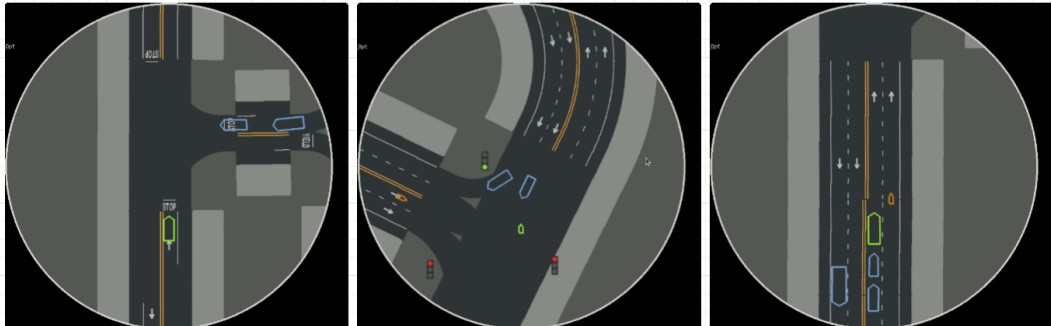

Figure 9: The **Urban Driving** environment characterizes a diverse range of traffic scenarios including intersections, stop sign, multilanes. It also involve complex interactions with other traffic agents and challenge the LLM agent to correctly recognize other agents and make reasonable decision-making to safely navigate in the environment.

### A.4.1 AGENT WORLD

We first introduce the combat rules, which determines the logic to calculate the outcome of each combat. The warrior (and the boss) has three states: attack, defense, and life. When a warrior with attack $a$, defense $d$, and life $l$ is fighting a monster with $a_m$, $d_m$ and $l_m$, we use following equation to calculate the outcome, which is the residual life of the warrior $l_{updated}$ after the combat:

$$turn = \frac{l_m}{a - d_m} \tag{3}$$

$$l_{updated} = l - threshold(a_m - d) * turn \tag{4}$$

If $turn < 0$, the warrior cannot attack the monster. If $l_{updated} < 0$, the warrior is defeated and we deem this as failure. This basic rule applies to both the imperfect environment and the perfect environment.

**Monsters.** The perfect world model also features a wider variety of monsters, categorized into three types:

- Normal: Balanced attack and defense values.
- Wizard: High attack but low defense.
- Guardian: High defense but low attack.

The imperfect world model only includes monsters of the first type.

**Store.** A unique feature of the perfect environment where the warrior can trade combat experience (gained from fighting monsters) to upgrade attack, defense, or life stats. The player must maintain a balanced approach to these upgrades. Imperfect world model does not have stores, or the concept of the combat experiences.

**Complexity** The imperfect environment has smaller rooms size, room numbers and simpler connectivity, while the perfect environment is more complex, and involves exploring different maps by using the *stairs*. For example, if we enter the stair in the upmost stairs of the first image in Figure 8, we will get the second floor, which is the second image in Figure 8. The imperfect environment only has one map to explore.

**Items.** In the imperfect world model, items are limited to swords and shields, which boost attack and defense, respectively. The perfect world model adds more strategic items, like the shovel, which can break a wall block, and the Wing, which allows the warrior to fly to the symmetrical position on the map. Using these items effectively requires a deep understanding of the game.

**Vampire.**    A special monster in the perfect environment with a unique lifesteal effect, which drains a fixed percentage of the warrior's life with each turn in combat. A common strategy for facing this monster is to keep the warrior's life relatively low.

**Keys & Doors.**    In the imperfect world model, there is only one type of key and door. In contrast, the perfect world model introduces keys and doors in three colors: yellow, blue, and red. The red key is the rarest, while the yellow key is the most common. For example, in the fourth image of Figure 8, to enter the middle room on the right, you can use either the blue door or the yellow door. The varying scarcity of these keys and the cost of defeating the guardian monsters present unique trade-off challenges for LLMs in making optimal decisions.

A.4.2    URBAN DRIVING

**Traffic Density.**    In the imperfect environment, traffic density is kept constant with a moderate number of vehicles on the road. In the perfect environment, traffic density varies dynamically, ranging from light to heavy congestion, which requires the driving model to adapt to different traffic flow patterns and congestion levels.

**Obstacle Types.**    The imperfect environment have only one obstacle, the fixed size vehicle. The perfect environment adds static obstacles such as parked vehicle and traffic cones. It also involves dynamic obstacles such as moving pedestrians, cyclists, and non-compliant behaviors such as vehicles suddenly stopping or swerving, which requires accurate responses from the driving model.

**Road Types.**    The imperfect environment consists solely of straight highways and simple turns. The perfect environment introduces a variety of road types, including sharp curves, urban streets with narrow lanes, lane merging, and complex intersections with traffic lights and STOP signs. Each road type presents unique challenges for the driving model.

**Interaction with Other Vehicles.**    In the imperfect environment, interactions with other vehicles are minimal and predictable, with other vehicles maintaining consistent speeds and distances. In the perfect environment, other vehicles exhibit more complex behaviors, such as sudden lane changes, erratic driving, and varied speed patterns, requiring the model to predict and react to these unpredictable actions effectively.

