# OpenReview forum: "GLIMO: Grounding Large Language Models With Imperfect World Models"
_ICLR.cc/2025/Conference — ICLR 2025 Conference Withdrawn Submission_

### Official Review · Reviewer_Yanc · 2024-10-28

**Soundness:** 2
**Presentation:** 2
**Contribution:** 2
**Rating:** 3
**Confidence:** 4

**Summary:**

This paper proposes using simulated environments to collect data for fine-tuning LLM for some decision-making tasks. The data collection part is also based on LLM, where humans provide three kinds of promptings/tools: RAG, Reasoning, and Exploration to ensure the data collection covers a decent distribution. The main claim is that by using extra collected synthetic data, the agent could perform better than using all human-annotated data. The authors verify their proposed approach in two kinds of environments (but all text-based).

**Strengths:**

- Show a possibly feasible way of generating massive data to fine-tune LLMs for "real-world" tasks.

- The use of RAG, Reasoning, and Exploration seems interesting.

**Weaknesses:**

- The first intriguing point is from Figure 1 about the definition of "world model". I think from Figure 1 the authors treat human annotation as a perfect world model and some simulated environments as an imperfect world model. I would definitely support that human intelligence is still better than current LLMs for sim or real decision-making tasks, but I don't think human-annotated data compose a true world model.

- I think the general high-level idea was already there tens of years ago when RL agents started to learn in simulated environments. This paper is about automating it using all LLMs.

- This paper is motivated by the cost and safety issue in collecting data in the real world but does not show any experiments on transferring from sim to real. (L49-73)

- The paper is only focusing on text-based LLMs (L238), which extremely limited the use case of the proposed method, since the visual world could be much more complex than text-based environments, and most of the real-world decision-making tasks require visual modality. Another important issue is that this paper is about **grounding** LLMs. With only text inputs/outputs, it's hard to say the model is grounded in some real applications.

- The data generation highly relies on the performance of the LLM used. Does it mean your approach is limited by how well GPT-4 "grounds" to the world model? I think an interesting point is to use your fine-tuned LLM to iteratively collect new data and see if it can outperform off-the-shelf models. More generally speaking, tune your LLM in the exact RL fashion / do online learning given the simulated environment.

- Experimental details are highly unclear. We need to know the amount of data, what types of tasks, and the definition of out-of-distribution tasks for Agentworld.

- Baselines are only base models rather than previous approaches. In its current shape, it's extremely hard to evaluate how significant of the proposed method is.

- Misc:
 * L16 trining -> training

**Questions:**

Please see the weaknesses part above.

---

### Official Review · Reviewer_MSYq · 2024-11-02

**Soundness:** 2
**Presentation:** 2
**Contribution:** 3
**Rating:** 5
**Confidence:** 3

**Summary:**

This paper introduces GLIMO, an automated LLM-guided dataset generation method to improve LLM's capability in embodied tasks. The designed generator assumes an imperfect world model on an environment, and incorporate three components: an interactive self-refining module, a question-answering instruction seed datasets, and a retrieval-augmented generation module to interact with the "simulator" and generate dataset.

With the dataset generated by GPT-4, the authors are able to fine-tune a list of open-sourced LLMs including LLaMA and OPT to perform better on 2 benchmarks: a grid-world strategy game called Agent World and a simulated self-driving environment Urban Driving.

**Strengths:**

- This paper introduces an interesting topic: instead of assuming a perfect world model, which is a hard assumption for many real-world tasks, it proposes to collect dates with more accessible and low-cost imperfect world models.
- The authors combine different prompting strategies to help with reasoning, exploration, and self-refinement of LLM agent.
- With the automatically generated dataset, a variety of LLMs are able to perform better on the domains the authors present.

**Weaknesses:**

- The evaluation of the "imperfect world model" is not comprehensive.
  - The authors claim to apply imperfect world model of an environment to gather dataset. However the imperfect world models (simulators) tested in the paper are a bit confusing. For example, for the Agentworld imperfect world model, instead of saying imperfect, it may be just irrelevant because it shows different elements and tools. Therefore it does not provide a "wrong" dynamics of the environment, but some irrelevant ones. It would be great if the authors can also simulate/test some other cases, for example, adding random noise to the simulators when collecting the data, but removing it for testing, which is more similar to the case of training of simulated environment before transferring to real-world applications.
- The method overlooks the potential model-specific mistakes.
  - The proposed method refines the policy of the data collection LLM, however, as the target goal is to improve a target LLM (for example, LLaMA). There can be some mistakes/inability that will commonly happen for LLaMA but not captured by GPT4. Therefore it may also help to consider the feedbacks from the target LLM themselves.

**Questions:**

- For Figure 5, the authors claim that the data from the imperfect world model can serve as an effective regularization technique. However, I wonder if it is not the data from the imperfect world model, but only mixing the data with real simulators to some random questions with the same format, how would the performance be?
- L226-227, the LLM agent iteratively executes a plan and evaluates whether the defined goal has been achieved or not. Is it a self-evaluation of the LLMs or feedback from the environment itself?
- What is the size of the final dataset? How do you determine when to stop the data collection?
- How much would the surprising score on L296 help with the performance?
- L245 what kind of human prior knowledge is integrated? Could you please give an example?

---

### Official Review · Reviewer_PiH4 · 2024-11-03

**Soundness:** 2
**Presentation:** 2
**Contribution:** 1
**Rating:** 1
**Confidence:** 5

**Summary:**

The paper introduces a study of finetuning large language models from the trajectory data in driving and game simulators.
The trajectory data is augmented using various methods, such as complex prompt template and QA data generated from previous experiences.
The authors shows the finetuning such LLMs on simulator generated have a 2-3x improvement on performance in task success rate and QA compared to the base models.

**Strengths:**

The authors finetuned a 70B model and reported its finding, which is quite difficult computation-wise.

**Weaknesses:**

I appreciate the authors spending time in fine-tuning large models, which is an interesting contribution on its own. However, I believe there are key weaknesses in this work that makes it unacceptable to be published at ICLR.

The authors attempt to validate the widely known fact that LLMs can excel in tasks such as 2D gaming and self-driving when provided with domain-specific data and prompt engineering. It is well-established from multiple works that LLMs benefit from past experiences (1 and related works), retrieval-augmented generation (numerous works), and self-refinement (2 and related works). The findings presented in this paper offer no additional insights beyond what is already known to the community.

The authors frame their paper as learning from an imperfect "world model," where the imperfect world model is a simulator. While I agree that a simulator can be considered an imperfect world model, the framing in the introduction, which suggests that learning from a simulator is "novel," demonstrates a lack of understanding and appreciation for the extensive work done by the robotics community on simulators.

Moreover, the modeling choice made by the authors (i.e., evaluating only language-based LLMs) significantly limits the impact and applicability of their method in relevant scenarios. The primary bottleneck in using LLMs for multimodal scenarios lies in representing the environment, a problem that the authors conveniently overlook.

The choice of environment also lacks a strong rationale. Although the authors cite robotics applications as their inspiration, no robotics-related applications are presented in the paper. I suspect that the authors omitted robotics simulators because it is challenging to obtain ground truth textual descriptions of the environment. However, addressing this problem is essential for anyone intending to work on robotics.

This work would have been much more valuable if the authors had focused on what they aimed to achieve (i.e., robotics environments), used multimodal foundation models, and carefully considered how different data choices affect model performance. In a data-driven world, providing insights into the types of data needed for various purposes in a robotics context would be highly beneficial. Simply applying well-established LLM methodologies falls short of ICLR standards.

1. Reflexion: Language Agents with Verbal Reinforcement Learning, Shinn et al.
2. SELF-REFINE: Iterative Refinement with Self-Feedback, Madaan et al.

**Questions:**

Q. Why do the authors ignore the problem of representing the environment as text?
Q. Why did the authors make a choice of a 2D game and a driving simulator when the authors claim to work in robotics?
Q. Why do authors ignore tons of work on learning in simulators and present the work as something novel?

---

### Official Review · Reviewer_xZyN · 2024-11-04

**Soundness:** 2
**Presentation:** 2
**Contribution:** 2
**Rating:** 3
**Confidence:** 3

**Summary:**

This paper proposes GLIMO, a pipeline that generates imperfect world data for improving the performance of executing robotic tasks. Specifically, this paper proposes to use LLM as the data generator to automatically create a high-quality and diverse instruction datasets. Specifically, in the data creation process, the LLM will iteratively self-refine itself, generate rationales and reasonings, and encourage exploration for better diversity. Empirically, this paper shows better performance compared with other zero-shot agents. Besides, this paper also shows some interesting findings like how the imperfect world data helps better generalization to out-of-distribution tasks.

**Strengths:**

1. The idea that utilizes LLM as the data generator for collecting imperfect world data to enhance models' performance is interesting and could be generalizable to more tasks that require physical understanding or world knowledge.
2. The data collection process sounds reasonable, which improves the base agent with some useful techniques in LLM prompt (e.g., RAG, self-consistency).
3. Empirical results demonstrate that learning from this data achieves better performance than other zero-shot baselines (e.g., a fine-tuned 70B model could achieve similar performance as GPT4).

**Weaknesses:**

1. It seems that a stronger LLM is utilized as the data generator (e.g., gpt4o L356-L358). Then, it seems that it becomes a distillation of a small LLM with the output from the stronger LLM, which in this case the improvement seems to be intuitive.
2. When comparing with other zero-shot baselines (e.g., GPT4), are all the techniques used for data annotation (e.g., RAG, self-consistency) also utilized for the zero-shot prompting? If including zero-shot prompting techniques could bring better performance, the contribution of distillation of a smaller/weaker LLM seems to be trivial.
3. Besides comparing with zero-shot GPT4 agent, what're some previous SoTA results on these benchmarks with LLMs?

**Questions:**

Please refer to weakness.

---

### Note · Authors · 2024-12-02

I have read and agree with the venue's withdrawal policy on behalf of myself and my co-authors.